# Expression, Characterization and Selective Chemical Inhibition of Essential *Schistosoma mansoni* Tegumental Acetylcholinesterase (SmTAChE)

**DOI:** 10.3390/ijms26051975

**Published:** 2025-02-25

**Authors:** Patrick J. Skelly, Akram A. Da’dara

**Affiliations:** Molecular Helminthology Laboratory, Department of Infectious Disease and Global Health, Cummings School of Veterinary Medicine, Tufts University, North Grafton, MA 01536, USA; patrick.skelly@tufts.edu

**Keywords:** schistosomiasis, *Schistosoma*, tegument, acetylcholinesterase, non-neuronal, therapy, enzyme inhibition

## Abstract

Schistosomiasis, a parasitic disease caused by worms of the genus *Schistosoma*, affects >250 million people worldwide. With no available vaccine, treatment relies solely on one drug—praziquantel—underscoring the urgent need for new therapies. We identified a tegumental, non-neuronal acetylcholinesterase (AChE) from *Schistosoma mansoni*—SmTAChE—as a promising drug target. RNA interference confirmed its essential role in parasite survival, as gene suppression significantly reduced parasite recovery from infected animals. Here, we produced functionally active recombinant SmTAChE by using a mammalian expression system. Biochemical characterization confirmed its identity as a true acetylcholinesterase, with the highest turnover rate (K_cat_ = 373 ± 39 s^−1^) and catalytic efficiency (K_cat_/K_m_ = 1.17 × 10^6^ M^−1^·S^−1^) for acetylthiocholine. Additionally, rSmTAChE was inhibited by classical AChE-specific inhibitors but not by a butyrylcholinesterase-specific inhibitor. To identify novel SmTAChE inhibitors, we developed a high-throughput chemical screening protocol (Z′ factor > 0.9) and screened a 1894-compound validation library. Twelve compounds reproducibly inhibited rSmTAChE by >30% at 7.5 µM, including known AChE inhibitors like physostigmine and new selective inhibitors. Notably, compound #2 preferentially inhibited rSmTAChE (IC_50_ = 0.74 µM) over human AChE (IC_50_ = 151 µM), thus providing a foundation for developing parasite-specific therapies targeting SmTAChE and potentially leading to new treatments for schistosomiasis.

## 1. Introduction

Schistosomiasis is an important parasitic disease caused by helminth parasites of the genus *Schistosoma*, affecting more than 250 million people worldwide; over 800 million people live at risk of infection [1,2,3,4,5,6]. Mortality is estimated at more than 280,000 deaths per year, with tens of millions having chronic morbidity [7,8]. For pregnant women, infection can result in poor fetal outcomes [7], and infected children can exhibit stunted physical and cognitive development [4]. There are three major species that cause schistosomiasis in humans: *Schistosoma mansoni*, *S. haematobium* and *S. japonicum*. Schistosomiasis arises following infection with larval parasites (cercariae), which emerge from infected freshwater snails. In the skin of a human host, the cercariae transform into juvenile forms called schistosomula, and these invade the bloodstream and mature into adult worms commonly called blood flukes. There is no vaccine to prevent schistosome infection. Control is essentially limited to treatment with praziquantel (PZQ) [9]. While safe and effective, this drug does not kill juvenile parasites and reinfection following treatment is common [10,11,12,13]. In addition, PZQ has been in wide use for over 40 years, and a series of laboratory studies and clinical trials have raised concerns about the development of tolerance and/or resistance to it [14,15,16,17,18]. The World Health Organization, recognizing that reliance on a single curative drug is risky, has called—together with others in the scientific community—for the urgent development of novel interventions for the prevention and cure of schistosomiasis [19,20]. The aim of this work is to develop one such intervention targeting schistosome acetylcholinesterase (AChE; EC. 3.1.1.7).

It has long been known that human schistosomiasis can be successfully treated by using chemicals that block AChE activity. For instance, the drug metrifonate, an organophosphorus compound [21,22] is metabolized to dichlorvos (2,2-dichlorovinyl dimethyl phosphate), and this acts by inhibiting AChE activity [23,24]. In schistosomes, in addition to the expected neuronal AChE activity, AChE activity has unexpectedly also been shown to be associated with the tegument (skin) of the intravascular worms [25,26,27,28]. Intact living worms can cleave exogenous acetylcholine [29]. It was suggested that the tegumental AChE is the target for metrifonate therapy [30]. Metrifonate is no longer commercially available as an anti-schistosome agent; it has reduced efficacy compared with PZQ, needs to be delivered in multiple doses, and has high specificity to human AChE, which results in problematic toxicity [21,31]. Despite this, the fact that AChE inhibition has been used to successfully treat schistosomiasis reinforces the notion that the identification of novel, safe, schistosome-specific AChE inhibitors deserves renewed attention.

Originally, it was hypothesized that a single gene encoded both the tegumental and neuronal forms of AChE in schistosomes [32,33]. However, we demonstrated that separate genes are involved; the SmAChE1 gene encodes the neuronal (internal) enzyme, while a second gene called SmTAChE encodes the (external-facing) non-neuronal AChE [29]. The sequences of the two proteins are quite distinct, with SmTAChE exhibiting just 36% amino acid identity with SmAChE1. SmTAChE, our focus in this work, contains all amino acids considered essential to AChE activity; this includes the catalytic triad (S^239^, E^401^, and H^553^), the choline-binding site (W^115^), six cysteines responsible for three intrachain disulfide bonds, four charged residues involved in forming two salt bridges and the strong conservation of residues lining the catalytic gorge [29]. The 694-amino acid SmTAChE enzyme also contains a predicted signal peptide (M^1^-S^25^) and has been shown experimentally to be glycosylphosphatidylinositol (GPI)-anchored [29]. In schistosomula and adult parasites, SmTAChE immunolocalizes to the tegument, as expected. Localization by immunogold electron microscopy further confirms that the protein is distributed in adult worm tegumental membranes [29].

We have demonstrated that the tegumental AChE performs an essential function in the worms in vivo; parasites whose SmTAChE gene is suppressed cannot establish robust infection in experimental animals [29]. Chemicals that mimic the RNAi effect should likewise debilitate the parasites, and our aim here is to identify such compounds. Finally, since SmTAChE and human AChE are quite distinct (exhibiting just ~35% sequence identity, including in the catalytic gorge), it is very likely that drugs preferentially inhibiting the schistosome enzyme will be identified. What important function SmTAChE fulfills in intravascular schistosomes is not known. Since acetylcholine can act as a vasodilator, SmTAChE might influence host vascular physiology and permit unobstructed movement of worms in the blood vessels. In addition, since several immune cells respond to acetylcholine, the enzyme may have an immunoregulatory role [34,35,36]. For example, it has been reported that T cells, B cells and macrophages express both muscarinic (mAChR) and nicotinic (nAChR) acetylcholine receptors and that the stimulation of these receptors causes various functional and biochemical changes in immune cells [35,37,38,39,40]. SmTAChE activity could dampen acetylcholine-driven immune signaling in the local environment of the worms.

In this study, we produced and functionally characterized SmTAChE in a mammalian expression system. Additionally, we performed a pilot screen of chemical compounds at the Broad Institute of the MIT and Harvard by using a high-throughput screen (HTS) validation chemical library. As reported here, this assay validation screen identified several SmTAChE inhibitors that, in the longer term, could form the basis of a new therapy to treat schistosomiasis.

## 2. Results

### 2.1. Characterization of rSmTAChE

To functionally characterize the tegumental AChE of *S. mansoni* (SmTAChE), we produced the enzyme in recombinant form by using suspension Chinese hamster ovary cells (CHO-S). Codon optimization significantly improved the expression of SmTAChE in CHO-S cells. This is clearly indicated by higher AChE enzyme activity in equivalent aliquots of medium from cells transfected with plasmid encoding the codon-optimized protein (AChE-Opt) compared with medium from cells transfected with a non-codon-optimized plasmid (AChE) (Appendix A). As expected, supernatants from control cells transfected with the empty plasmid do not display AChE activity at any time (Appendix A, Control). Western blotting using anti-SmTAChE antibodies confirms more robust rSmTAChE production in the cells transfected with the codon-optimized plasmid (arrow, lower panel, Appendix A). While considerable amounts of rSmTAChE could be produced by using transiently transfected cells, to simplify production, we generated several stable cell lines that were able to routinely produce 5–10 mg of purified recombinant protein per liter of culture supernatant under standard culture conditions. The recombinant enzyme was purified to homogeneity by IMAC using Excel Nickle chromatography. Figure 1A shows an aliquot of purified rSmTAChE resolved by SDS-PAGE and stained with BioSafe Coomassie stain (left panel, arrow). Since the estimated molecular mass of the purified protein on the gel was larger than its predicted mass, we hypothesized that the recombinant protein was glycosylated; the SmTAChE sequence contains five potential N-glycosylation sites [29]. To test this, we treated an aliquot of the purified protein with deglycosidase (PNGase F) and then resolved this sample by SDS-PAGE and subjected it to Western blot analysis. As shown in Figure 1A (right panel), treatment of rSmTAChE with PNGase F (+ lane) resulted in a shift in the molecular weight of the enzyme to about its predicted size (~75 kDa, compared with that seen when PNGaseF is not used, − lane).

To evaluate the substrate specificity of rSmTAChE, we measured its catalytic activity by using acetylthiocholine (ATCh), butyrylthiocholine (BuTCh) and propionylthiocholine (PTCh) as substrates. As shown in Figure 1B, the recombinant enzyme is specific to ATCh. Very weak activity was measured when PTCh was used as a substrate, and almost no activity was detected when BuTCh was used. The recombinant SmTAChE has a K_m_ value of 0.32 ± 0.04 mM for acetylthiocholine; the Michaelis–Menten curve is presented in Figure 1C. This K_m_ is comparable to that for BuTCh (0.33 ± 0.02 mM) and 2.5 times higher than that for PTCh (0.14 ± 0.002 mM) (Table 1). However, the turnover rate (K_cat_) for ATCh (372.79 ± 38.6 S^−1^) is 35-fold higher than that for BuTCh (10.67 ± 0.22 S^−1^) and 5-fold higher than that for PTCh (74.16 ± 0.35 S^−1^) (Table 1) (ANOVA, *p* < 0.0001). In addition, rSmTAChE has significantly higher catalytic efficiency (K_cat_/k_m_) for ATCh (1.17 ± 0.14 × 10^6^ M^−1^·S^−1^) than the other two substrates (ANOVA, *p* < 0.002) (Table 1). Finally, we determined the K_m_ value for native AChE on the tegumental surface of live parasites. As for rSmTAChE, a similar K_m_ value was obtained for native AChE on the surface of living schistosomula (K_m_: 0.39 ± 0.06 mM; Figure 1D).

### 2.2. Inhibition of rSmTAChE with Known AChE Inhibitors

Next, we performed inhibition experiments by using several established AChE-specific inhibitors, as well as the butyrylcholinesterase-specific inhibitor tetraisopropyl pyrophosphoramide (iso-OMPA). As shown in Figure 2 (red line), iso-OMPA displayed no appreciable inhibition of rSmTAChE, except at very high concentrations. On the other hand, known AChE inhibitors easily inhibited rSmTAChE, with IC_50_ values ranging from 0.1 to 100 µM (Figure 2, Table 2). The most rSmTAChE inhibitory compounds tested here were dichlorvos and physostigmine, with IC_50_ of 0.12 µM and 0.37 µM, respectively. Furthermore, we evaluated the inhibitory activity of these inhibitors on tegumental AChE activity in live schistosomula parasites. As shown in Table 2, these inhibitors had broadly similar IC_50_ values for rSmTAChE compared to that displayed by the native enzyme in live parasites. No AChE inhibition was observed in the live parasite assays when iso-OMPA was tested.

### 2.3. Screening of MLPCN-Broad Institute Validation Chemical Library

To facilitate a high-throughput screen for potential inhibitors, the AChE assay was modified for use in a 384-well plate format. This assay was found to have a Z′ factor of 0.93, an S/B ratio of 16.6 and an S/N ratio of 51.6, suggesting that the assay is robust and suitable for use in high-throughput chemical screening experiments. By using this assay, we undertook a screen of 1894 compounds by using the MLPCN-Broad Institute Validation Library set of compounds. The Library was screened in duplicate on two different days to assess assay performance and reproducibility. Both screens had an average Z′ of 0.90 for all plates. Figure 3 shows the combined duplicate screen results, where neutral control samples (containing no screening compounds) are depicted in light gray, inhibitor control samples (containing no enzyme) are in red and test compound samples are in blue. The red dashed line depicts the cutoff: 3 standard deviations from the mean control value, as well as more than 30% inhibition of enzyme activity. The two replicates are plotted against each other and show a linear fit of the data set. In almost all cases, duplicate values are very close, showing good day-to-day assay reproducibility. In the lower portion of the figure, bounded by the dashed red line, there are 12 compounds of interest, since they reproducibly and substantially inhibit rSmTAChE in this assay. The 12 compounds are listed in Table 3, from most to least inhibitory. Among the most potent of the 12 inhibitors are well known AChE inhibitors like physostigmine (eserine) (compound #1) and two salt forms of its analog, neostigmine (compounds #3 and #4). However, novel inhibitors were also identified, and these include compound #2 ([2-(2-fluorophenyl)ethyl]{3-methoxy-4-[2-oxo-2-(1-piperidinyl)ethoxy]benzyl}amine hydrochloride), which resulted in 85% inhibition of rSmTAChE, and compound #5 (Pentamidine isethionate salt), which resulted in 69% inhibition of rSmTAChE (Table 3). Compounds #8–12 exhibit variable ability to inhibit rSmTAChE (33–49%).

#### Specificity of Selected New rSmTAChE Inhibitors

Compound #2 ([2-(2-fluorophenyl)ethyl]{3-methoxy-4-[2-oxo-2-(1-piperidinyl)ethoxy] benzyl} amine hydrochloride) is identified here for the first time as a potent AChE inhibitor, with no other previous biological activities reported. Therefore, we analyzed the specificity of this compound against rSmTAChE versus recombinant human AChE (rHsAChE). IC_50_ plots of compound #2 for both enzymes are presented in Figure 4A. Compound #2 is seen to have a ~200-fold lower IC_50_ value for schistosome rSmTAChE (0.76 µM) compared with human AChE (151 µM). On the other hand, as shown in Figure 4B, physostigmine, a previously known AChE inhibitor, is more specific to the human enzyme (IC_50_ = 0.02 µM) than to the schistosome enzyme (IC_50_ = 0.134 to 0.36 µM). To further explore compound #2 as an SmTAChE-specific inhibitor, we obtained two commercially available analogs (one is a 2D image of compound #2 with 90% similarity, and the other is a 3D image of compound #2 with 94% similarity (ChemBridge Corp., San Diego, CA, USA)). The 2D analog ([2-(3,4-dimethoxyphenyl)ethyl]{3-methoxy-4-[2-oxo-2-(1-piperidinyl)ethoxy]benzyl} amine hydrochloride) was not as strong as an inhibitor of AChE, with an IC_50_ of 30.67 µM for rSmTAChE (40 times higher than the parental compound #2) and an IC_50_ of 24.32 µM for rHsAChE (Figure 4C). The 3D analog (N-{3-methoxy-4-[2-oxo-2-(1-piperidinyl)ethoxy]benzyl} cyclooctanamine hydrochloride) had an IC_50_ of 5.71 µM for rSmTAChE and an IC_50_ of 17.47 µM for rHsAChE (Figure 4D). To further characterize compound #2, we obtained 25 additional analogs of compound #2 (commercially synthesized at WuXi Apptec Co., Ltd., Tianjin, China). These compounds inhibited rSmTAChE to varying levels (Appendix A); however, none of the new analogs was more potent than the original compound #2.

Compound #8 (3-(2,3-dihydro-1,4-benzodioxin-6-yl)-6-ethyl-7-hydroxy-8-[(2-methyl-1-piperidinyl) methyl]-4H-chromen-4-one) was also identified here for the first time as an AChE inhibitor; it inhibited rSmTAChE by ~50% in the initial screen (Table 3). Dose–response analysis shows that it inhibits rSmTAChE with an IC_50_ of 5.45 µM; in contrast, the IC_50_ value for the human enzyme was >200 µM (Figure 5). Compounds 5, 6, 7 and 9–12 were also identified as inhibitors of rSmTAChE, but these were not further tested in this work.

## 3. Discussion

Schistosomiasis is a serious global public health problem [3,4] for which no vaccine is currently available. Control is essentially limited to treatment with a single drug—praziquantel (PZQ)—and new treatment options are needed. In this paper, we investigate schistosome tegumental acetylcholinesterase (SmTAChE) as a rational target for new anti-schistosome therapeutics. SmTAChE is highly expressed in all intravascular life stages and is exposed on the external surface of the worm’s tegument [29]. Furthermore, knocking down SmTAChE activity by using RNAi significantly impairs the ability of parasites to establish infection in mice, showing that SmTAChE performs an essential function in the worms in vivo [29]. Given this, it is likely that chemicals that block SmTAChE function, as sought here, likewise debilitate the worms. Finally, AChEs as an enzyme class are known druggable targets; AChE inhibitors have been used in the treatment of various human diseases, such as myasthenia gravis, Alzheimer’s disease, postoperative ileus, bladder distention and glaucoma [41,42,43,44,45].

To begin our work, we first cloned and expressed SmTAChE as a recombinant protein in a eukaryotic expression system (CHO-S cells). Production was greatly enhanced by driving protein expression from a plasmid that had been codon-optimized for synthesis in hamster (CHO) cells. Exposure of the protein to glycosidase (PNGase F) lowers its molecular mass, showing that the enzyme is a glycoprotein. To characterize the biochemical nature of SmTAChE, we measured its kinetic characteristics with the substrates acetylthiocholine (ATCh), butyrylthiocholine (BuTCh) and propionylthiocholine (PTCh). AChEs preferentially hydrolyze ATCh, followed by PTCh, with little to no hydrolysis of BuTCh, while butyrylcholinesterases, as their name suggests, prefer BuTCh as a substrate [46]. SmTAChE was found to have a strong preference for ATCh, with significantly higher turnover (K_cat_) and catalytic efficiency (K_cat_/k_m_) for ATCh versus the other two substrates. In addition, specificity was substantiated by demonstrating its selective inhibition by BW284c5.1, a highly specific acetylcholinesterase inhibitor, and the lack of inhibition when iso-OMPA, a butyrylcholinesterase-specific inhibitor, was used [47,48]. These findings allow us to classify SmTAChE as a “true” acetylcholinesterase.

The kinetics of SmTAChE action were very similar whether measured by using the recombinant enzyme or the native enzyme displayed on the surface of living parasites, suggesting that the recombinant enzyme has a similar conformation to the native enzyme. In addition, IC_50_ values obtained by using known AChE inhibitors were quite similar as assessed by using either rSmTAChE or live worms, suggesting that the native enzyme, like its recombinant counterpart, is accessible for chemical inhibition. These findings provide confidence that any new chemicals identified in this work that block recombinant enzyme function will be able to similarly impede the action of the native enzyme in worms.

Armed with a recombinant enzyme that faithfully reflects the native SmTAChE enzyme on live schistosomes, we set out to identify specific inhibitors of rSmTAChE by looking for the ability of a collection of 1894 compounds in the MLPCN-Broad Institute Validation Chemical Library to block rSmTAChE enzyme activity. Our activity assay, miniaturized for automation in a 384-well format, yielded a robust Z′ score (0.9) and led to the identification of a set of 12 compounds that reproducibly inhibited rSmTAChE activity by at least 30%. Encouragingly, several of these compounds are already known acetylcholinesterase inhibitors—thus validating the assay; for instance, physostigmine salicylate (compound #1) and two salts of its analog, neostigmine methyl sulphate (compound #3) and neostigmine bromide (compound #4), are all potent SmTAChE inhibitors in our assay. Despite its associated toxicity, physostigmine is widely used in the treatment of myasthenia gravis [41] and Alzheimer’s disease [49,50]. However, due to adverse side effects, further research into this drug has been no longer recommended [49,51]. Furthermore, we find that physostigmine is, like dichlorvos mentioned earlier, more specific to human AChE, with an IC_50_ ~20-fold more potent for human versus schistosome AChE [52].

SmTAChE exhibits only ~35% amino acid sequence identity with human AChE [29]. Exploiting differences between parasite versus host enzyme should permit us to identify parasite-specific inhibitors that have low/no inhibitory effects on host AChE. For instance, the active site of SmTAChE is predicted to have an additional 27 amino acids compared with human AChE [29]. This would result in a change in the size and the composition of the active gorge, which would theoretically allow larger inhibitors to more easily access the parasite enzyme’s active site. Furthermore, we found that unlike recombinant human AChE, which is inhibited by substrate concentrations higher than 1 mM [53,54], rSmTAChE remains active above a higher substrate concentration (>10 mM), further affirming structural differences between the two enzymes. Shafferman et al. demonstrated that mutations in D^74^ and W^286^ in human AChE resulted in a loss of substrate inhibition [54]. These residues are part of AChE’s peripheral anionic site. Interestingly, the analogous positions of these residues in SmTAChE are F^102^ and D^320^, respectively [29]—differences that likely explain the lack of substrate inhibition seen with SmTAChE.

Compound #2 ([2-(2-fluorophenyl)ethyl]{3-methoxy-4-[2-oxo-2-(1-piperidinyl)ethoxy] benzyl}amine hydrochloride) was identified in our assay and is our best lead molecule. It inhibited rSmTAChE activity by ~85% in our initial screen (second only to eserine) and was found to have a ~200-fold lower IC_50_ value for schistosome SmTAChE (0.76 µM) compared with human AChE. To try to generate compounds of greater potency, we synthesized 25 analogs of compound #2 and compared their IC_50_ values. Included in this analysis were two additional, off-the-shelf analogs (Comp. #2-2D and Comp. #2-3D structural analogs). None of these 27 total compounds yielded an improved IC_50_ value over that of the original compound #2 in enzyme inhibition assays. Nonetheless, further SAR studies have the potential to lead to new compounds with higher specificity for the parasite’s enzyme.

Additional hit chemicals include compounds #8 (3-(2,3-dihydro-1,4-benzodioxin-6-yl)-6-ethyl-7-hydroxy-8-[(2-methyl-1-piperidinyl)methyl]-4H-chromen-4-one) and #9 (6,7-Dimethyl-3a,4,5,9b-tetrahydro-3H-cyclopenta[c]quinoline-4-carboxylic acid). Both resulted in ~50% inhibition of rSmTAChE. These are screening compounds which have no previously known targets and were identified here for the first time as novel AChE inhibitors. Further analysis of compound #8 shows that it has an IC_50_ of 5.22 µM compared with an IC_50_ of >200 µM (the highest concentration used in the IC_50_ analysis) for rHsAChE. These data show that this approach can result in the successful identification of molecules that preferentially inhibit the parasite’s enzyme. Compound #9 was not able to inhibit rSmTAChE in a dose-dependent manner and was consequently not pursued further.

Compound #5 (Pentamidine isethionate) was found to inhibit rSmTAChE by ~69% in our initial screen. This compound is an organosulfonate salt that was originally developed as an anti-trypanosomal drug [55,56,57,58]. The compound is known to have both antiprotozoal and antifungal activity [59,60]. It is considered to be the safest of the diamidine group of antiprotozoals, although it still has significant toxicity [61]. The mechanism of action is unclear, but in trypanosomes, it appears to bind to DNA, interfering with DNA replication and function [58,62]. Here, we show for the first time that this compound inhibits AChE.

Additional hits include butacaine (compound #6), which exhibited 69% inhibition of rSmTAChE and is a known anesthetic that blocks nerve conduction. Butacaine was reported to be a phospholipase inhibitor [63] and to inhibit voltage-gated Na^+^ channels [64]. Here, we add AChE inhibitor to its known effects.

Interestingly, compound #7 (1-[(2-[diethylamino]ethyl)amino]-4-[hydroxymethyl]-9h-thioxanthen-9-one), more commonly known as hycanthone (HYC), is a drug that was formerly used to treat schistosomiasis [65,66]. HYC’s anti-schistosome activity was proposed to first involve its activation by sulfation, followed by the subsequent alkylation of worm macromolecules (such as DNA) to exert a lethal effect [67,68]. Our data suggest that inhibiting parasite AChE might directly contribute to the ability of HYC to kill schistosomes. Earlier studies have demonstrated that HYC can inhibit schistosome cholinesterase, while cholinesterase activity from mouse brain was not affected significantly by this drug [65,66]. Adverse properties associated with HYC use, including mutagenic, teratogenic and carcinogenic activities, coupled with the introduction of PZQ as a safer alternative, led to HYC being withdrawn as an anti-schistosome reagent [67]. Little information is available about the remaining major hit compounds identified here (#8–12); their ability to inhibit rSmTAChE varies from 33 to 49% in the reported screen, and further work is needed to advance their candidacy.

Since any new therapy for schistosomiasis should ideally target all three medically important species, we previously looked for evidence of surface AChE activity in *S. japonicum* and *S. haematobium*, in addition to *S. mansoni* [29]. Adult male and female worms of all three species cleaved exogenous acetylthiocholine, with *S. haematobium* parasites exhibiting the highest surface AChE activity, relative to the other two species. SmTAChE homologs were subsequently identified and cloned from *S. haematobium* and *S. japonicum*. These were designated as ShTAChE and SjTAChE, respectively. We targeted the predicted surface AChE gene from each species for suppression by RNAi. In each case, this yielded parasites with a significantly diminished ability to cleave exogenous acetylthiocholine compared with the controls, confirming that the identified genes do encode the tegumental AChEs in the three schistosome species [29]. All three schistosome AChEs are very similar in protein sequence; SmTAChE exhibits ~85% sequence identity with the *S. haematobium* enzyme ShTAChE and ~71% identity with *S. japonicum* SjTAChE. This makes it highly likely that any SmTAChE-specific inhibitors identified by our ongoing efforts will additionally block the surface AChEs of the other two medically important schistosome species and will perform as efficacious, non-toxic agents that act broadly against all human schistosomes.

## 4. Materials and Methods

### 4.1. Parasites

The Puerto Rican strain (NMRI) of *Schistosoma mansoni* was used. Schistosomula were prepared from cercariae released from infected snails as described previously [69]. All parasites were cultured in DMEM/F12 medium supplemented with 10% heat-inactivated fetal bovine serum, 200 µg/mL streptomycin, 200 U/mL penicillin, 1 µM serotonin, 0.2 µM Triiodo-l-thyronine and 8 µg/mL human insulin and were maintained at 37 °C, in an atmosphere of 5% CO_2_ [70].

### 4.2. Expression and Purification of rSmTAChE

The full coding sequence of SmTAChE (GenBank accession number OP018961), including the putative signal peptide and GPI anchor domains, was optimized with hamster codon preferences and synthesized commercially (GenScript) (Appendix A). The region coding for amino acids T^27^-W^661^ (i.e., lacking the signal peptide and GPI anchoring signal) was amplified by PCR from adult *S. mansoni* cDNA or from the codon-optimized sequence. Amplified PCR products were purified and cloned into the pSecTag2A plasmid (Invitrogen, Waltham, MA, USA) at the *Asc*I and *Xho*I sites in frame with the Igκ leader sequence at the 5’ end and Myc and 6-histidine tags at the 3’ end. Correct cloning was confirmed by sequencing the final plasmid construct (Genewiz, South Plainfield, NJ, USA). Plasmid DNA was purified by using an EndoFree Plasmid Maxi kit, according to the manufacturer’s instructions (Qiagen, Germantown, MD, USA), and quantified by using a NanoDrop 2000 spectrophotometer (ThermoFisher Sci., Waltham, MA, USA).

To express recombinant SmTAChE (rSmTAChE), suspension-adapted FreeStyle Chinese hamster ovary cells (CHO-S cells) were used (Invitrogen). CHO-S cells were grown in serum-free Freestyle Expression Medium supplemented with 8 mM L-glutamine (ThermoFisher Scientific) at 37 °C and 8% CO_2_ in vented culture flasks under shaking at 140 rpm. For transient transfection, 40 mL of culture medium was seeded with CHO-S cells at 0.5 million cells per ml. The next day, cell number and viability were evaluated by using the Trypan blue exclusion test [71]. Cells (at a density of ~1 million/mL) were transfected using FreeStyle Max transfecting reagent and highly pure plasmid DNA, according to the manufacturer’s instructions (Invitrogen). In most protocols, cells were transfected with 1 µg of plasmid DNA and 1 µL of FreeStyle Max reagent per million cells. Transfected cells were incubated as before, and samples were harvested at various time points to monitor cell viability and rSmTAChE production. Recombinant protein expression was assessed in culture supernatant aliquots by using our standard AChE activity assay, described below, and by Western blotting by using specific anti-rSmTAChE antibodies, produced as described [29]. Optimal recombinant protein expression coupled with high cell viability was detected 72 h after transfection. To facilitate protein production, stable cell line clones secreting rSmTAChE were generated as follows: Briefly, 48 h after transfection, cells were treated with zeocin (250 µg/mL) for three weeks. Medium was exchanged every 48 h. Individual clones were isolated by plating each stable cell line at a cell density of ~0.5 cell/well in 96 well plates. Individual clones were analyzed for rSmTAChE expression as described above.

Recombinant SmTAChE was purified from cell culture medium by standard Immobilized Metal Affinity Chromatography (IMAC) by using HisTrap™ Excel columns, as described by the manufacturer (GE Healthcare Life Sciences, Marlborough, MA, USA). Purified rSmAChE was dialyzed overnight at 4 °C against phosphate-buffered saline (PBS) and then concentrated by ultrafiltration centrifugation by using Pierce Protein Concentrators (10K MWCO, Thermo Fisher Scientific, Waltham, MA, USA). Protein purity was assessed by using SDS-PAGE, and protein concentration was determined by using a BCA Protein Assay Kit (Pierce, Rockford, IL, USA).

### 4.3. Deglycosylation of rSmAChE

The N-glycosylation status of rSmTAChE was determined by using peptide-N-Glycosidase F (PNGase F) according to the manufacturer’s instructions (New England Biolabs, Ipswich, MA, USA). Briefly, ~5 µg of rSmTAChE was denatured at 100 °C for 10 min in the presence of 0.5% SDS and 40 mM DTT. Then, NP-40 was added to 1%. Deglycosylation buffer and PNGase F were added, and the mixture was incubated at 37 °C for 3 h. Samples were resolved by SDS-PAGE and analyzed by Western blotting as described below.

### 4.4. Western Blot Analysis

PNGase F-treated and untreated control protein samples were resolved by 4–20% SDS-PAGE (BioRad, Hercules, CA, USA), transferred to a PVDF membrane and probed, as previously described [29]. Briefly, the membrane was blocked with TBST (Tris-buffered saline, pH 7.5, 0.05% Tween 20) containing 5% dry non-fat milk powder for 1 h at room temperature. The membrane was then washed and incubated with primary rabbit anti-SmTAChE antibody (1:1000) for 1 h at room temperature, followed by washing with TBST buffer for 30 min and incubation with horseradish peroxidase-labeled donkey anti-rabbit IgG (1:5000) (GE Healthcare Life Sciences, Marlborough MA, USA) for 1 h at room temperature. Signals were detected by using ECL Western Blotting Detection Reagents (GE Healthcare), and the membrane was exposed to X-ray film. Films were then scanned by using a Kodak Image Station 2000RT.

### 4.5. Characterization of SmTAChE

AChE activity was measured at room temperature (~25 °C) following the modified Ellman method, using acetylthiocholine iodide (ATCh; Sigma-Aldrich, St. Louis, MO, USA) as substrate [29,72]. Standard reactions were performed in 96-well plates, and the reaction mixture (200 µL total volume, in duplicates) contained 10 ng of rSmTAChE diluted in 100 µL of 100 mM sodium phosphate (pH 7.2) and 100 µL of substrate solution (1 mM ATCh and 1 mM 5,5′-dithiobis(2-nitrobenzoic acid) (DTNB) diluted in 100 mM sodium phosphate (pH 7.2)). Wells containing no enzyme (100 µL of buffer and 100 µL of substrate) served as blanks. Absorbance at 412 nm was monitored over 1 h by using a Synergy HT spectrophotometer (Bio-Tek Instruments, Winooski, VT, USA). In some enzyme assays, varying concentrations (indicated) of butyrylthiocholine iodide (BuTCh, Sigma) or propionylthiocholine iodide (PTCh, Sigma) were tested as substrates. Initial experiments were conducted by using culture supernatants from cells transfected with either the standard AChE sequence or the codon-optimized sequence as a source of secreted acetylcholinesterase. Samples were collected at different time points after transfection, and varying volumes were analyzed in triplicate, as described above (Appendix A).

Kinetic parameters were measured in 96-well plates by using 10 ng of purified rSmTAChE. This amount displays linear concentration dependency of enzyme activity (Appendix A). The concentration of DTNB was fixed at 1 mM, while ATCh (or BuTCh or PTCh) was used at varying concentrations (0.01 mM to 100 mM). Enzyme activity was monitored continuously over 1 h at OD 412 nm by using a Synergy HT spectrophotometer. Enzyme velocity (determined as milli-OD units/min, mOD/min) was plotted against substrate concentration. The velocity data (mOD/min) were converted into actual amount of product (μmol/min/assay) by using the molar extension coefficient for 2-nitro-5-thiobenzoic acid (NTB) of 14.15 mM^−1^ cm^−1^ at 412 nm [73,74]. Michaelis–Menten kinetic parameters (K_m_, V_max_, K_cat_ and K_cat_/K_m_) for each substrate were determined by fitting specific AChE activity and substrate concentration by using GraphPad Prism V10.4 (GraphPad Software, Boston, MA, USA).

To measure the AChE activity displayed by living parasites, groups of ~1000, 7-day old schistosomula were incubated at room temperature in replicate (at least 5 replicates per substrate concentration) in clear, serum-free DMEM/F12 medium containing 1 mM DTNB and varying concentrations of ATCh. Changes in absorbance at OD_412_ were monitored as above. The experiments were repeated at least three times. All parasites remained alive and active by the end of the assay.

### 4.6. Miniaturization of AChE Assay

In order to conduct a screen of potential schistosome tegumental acetylcholinesterase (SmTAChE) inhibitors, the AChE assay based on Ellman’s reagent was first modified to a 384-well plate format. The assay was performed in 100 mM sodium phosphate buffer, pH 7.2. The final reaction volume was 50 µL per well, containing 5 ng of rSmTAChE. Absorbance changes were detected at 412 nm by using a Synergy HT machine (Bio-TEK) or at 405 nm by using an Envision multimode plate reader (Perkin Elmer, Waltham, MA, USA). Samples were dispensed by using a Well-Mate plate dispenser (Matrix, Singapore). Assay quality was evaluated by utilizing the statistical parameters of screening window coefficient (Z′ factor), signal-to-background (S/B) and signal-to-noise (S/N) ratios [75] by using the following equations: Z′ = 1 − [(3 × SD_E_ + 3 × SD_c_)/|AV_E_ − AV_c_|], S/B = AV_E_/AV_c_, S/N = (AV_E_ − AV_c_)/(SD_E_^2^ + SD_c_^2^)^0.5^, where AV_E_ and SD_E_ correspond to the average and standard deviation of the signal in the presence of the AChE enzyme, and AV_c_ and SD_c_ correspond to the average and standard deviation of the signal in the absence of the enzyme.

### 4.7. Validation of SmTAChE Chemical Screen

rSmTAChE was first tested by using the MLPCN-Broad Validation Chemical Library, which was developed by the Broad Institute of the MIT and Harvard, Cambridge, MA, USA [76]. This small-molecule library contains a validation set of 1894 compounds which contains a range of structurally diverse small molecules known to inhibit a variety of protein families. The Library was screened in duplicate on two different days to assess variability. Five nanograms of rSmTAChE was used in each well. The enzyme was diluted in 100 mM sodium phosphate buffer (pH 7.2), and 30 µL was dispensed in 385-well plates (in columns 1 to 23). One set of wells lacked the enzyme (receiving 30 µL of buffer) and served as the 100% inhibition reference. Samples were distributed by using a 384 Thermo Combi dispenser with standard cassette. Screening compounds were transferred from the validation library plates to assay plates by using a Pintool Cybio with 100 nl pins, and the final compound concentration was set to 7.5 µM. No chemical compounds were added to a number of wells, which served as 0% inhibition controls. Additional controls included adding 100 nl of DMSO (final concentration: 0.2%) instead of test compounds. The assay plates were then incubated for 20 min, and the reactions were initiated by adding 20 µL of substrate solution to all wells. The final substrate concentration was 0.35 mM acetylthiocholine and 0.35 mM Ellman’s reagent (DTNB) in 100 mM sodium phosphate buffer, pH 7.2. This substrate concentration was chosen because it is roughly equivalent to the K_m_ value of rSmTAChE. Assay plates were then read at 405 nm twice, i.e., immediately after the addition of substrate and 20 min later, by using an Envision multimode plate reader. Screening data were analyzed by using Genedata Assay Analyzer 10.0.2 Standard. The data were normalized to compound wells. SciTegic Pipeline Pilot 7.0 was used to pair duplicate data points between the two validation runs prior to the creation of screening graphics with TIBCO Spotfire 3.3.1. Compounds resulting in >30% inhibition of enzyme activity in 2 independent screens were selected for further evaluation. The IC_50_ values for a subset of these compounds were determined as described next.

### 4.8. IC_50_ Determination

IC_50_ values were determined in 384-well plates by using a fixed amount of rSmTAChE (5 ng) and serially diluted inhibitors. Enzyme activity was determined as described above and monitored continuously at OD_412_ nm. As before, inhibitors, or the DMSO control, were incubated with the enzyme for 20 min before adding the substrate solution. The final concentration of DMSO in the assay was 0.2%, and preliminary analysis showed that this concentration by itself yielded a reduction of <5% in enzyme activity. The reactions were initiated by the addition of 20 µL of substrate solution to all wells, and the rate of each reaction was measured (in mOD/min). Residual enzyme activity (percentage vs. control) was also calculated and plotted against the log inhibitor concentration (Log_10_[I]). The IC_50_ for each inhibitor was obtained by nonlinear regression analysis of log[I] vs. normalized response–variable slope by using GraphPad Prism v. 10.4. For comparison, the IC_50_ values of some compounds were also determined for human AChE by using commercially available recombinant human AChE (Sigma) by using the same protocol.

### 4.9. Evaluation of SmTAChE Inhibitor Impact on Enzyme Activity of Live Parasites

To monitor the effects of selected compounds on native SmTAChE activity on living parasites, ~250 two-day-old schistosomula per well of a 384-well microtiter plate were pre-incubated, in quadruplicate, with varying concentrations of each test compound, or DMSO, for 10 min [29]. The assay was conducted in serum-free, phenol red-free RPMI culture medium (as for the buffer, this medium does not interfere with the assay either). Next, substrate and Ellman’s reagent were added, as described above. The plates were read immediately thereafter and every 5 min for 30 min at 412 nm, and the rate of each reaction was measured in mOD/min.

### 4.10. Statistical Analysis

Student’s *t*-test and one-way analysis of variance (ANOVA) with Tukey’s post hoc analysis were used to compare the means between a target group and a control group, and *p*-values less than 0.05 were considered significant. Data were assessed for normality by using Shapiro–Wilk tests with GraphPad Prism 10.4. Bartlett’s test for homogeneity of variances was used to confirm the assumption of equal variances.

## 5. Conclusions

In this study, we identified and characterized a molecularly defined target for schistosomiasis treatment, namely, tegumental, non-neuronal AChE (SmTAChE). We successfully produced the *S. mansoni* enzyme in large quantities, characterized its biochemical properties and validated a high-throughput screening strategy. Our screening protocol identified 12 compounds that reproducibly inhibited rSmTAChE activity by at least 30% at 7.5 µM. Among these, several (compounds 1, 3 and 4) are known AChE inhibitors; however, due to their known high relative inhibition of human AChE, they appear unsuitable for development as new anti-schistosome drugs. Several other anti-SmTAChE chemicals were identified here, with a particular interest in compound 2, a new and potent inhibitor that is highly specific to SmTAChE. This compound, along with other SmTAChE inhibitors discovered in this study, holds potential as a lead compound for the development of a new therapy for schistosomiasis.

## Figures and Tables

**Figure 1 ijms-26-01975-f001:**
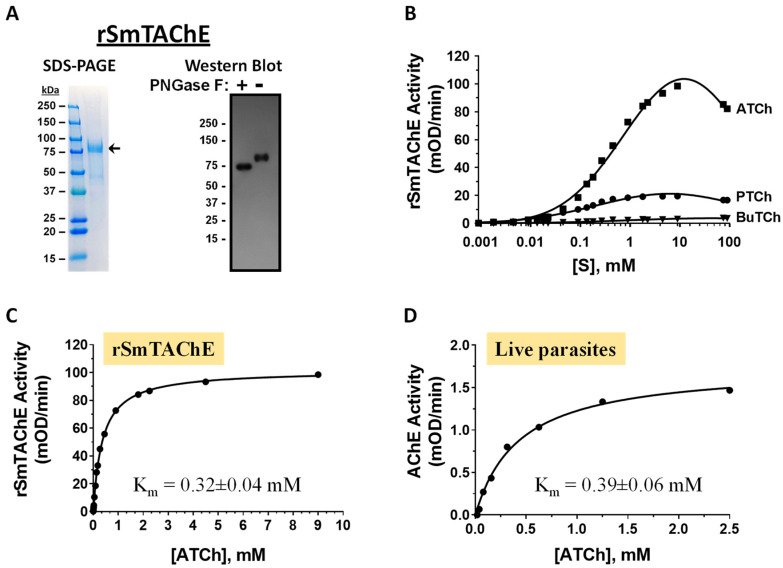
Characterization of SmTAChE. Recombinant SmTAChE (rSmTAChE) was purified from culture supernatant and analyzed by using SDS-PAGE (**A**, **left**); arrow shows purified rSmTAChE. The glycosylation status of rSmTAChE was analyzed by using PNGase F treatment and Western blotting (**A**, **right**). As shown, following PNGase F treatment (+), the molecular mass of rSmTAChE decreases, indicating that the enzyme is N-glycosylated. (**B**) Substrate specificity of rSmTAChE. The rate of hydrolysis (mOD/min) of acetylthiocholine (ATCh), propionylthiocholine (PTCh) and butyrylthiocholine (BuTCh) was determined for a range of substrate concentrations (1 µM to 100 mM). As shown, ATCh is the preferred substrate for the enzyme. Michaelis–Menten kinetics of rSmTAChE (**C**) and native enzyme on living schistosomula (**D**) are shown, with ATCh as a substrate. Km values are presented.

**Figure 2 ijms-26-01975-f002:**
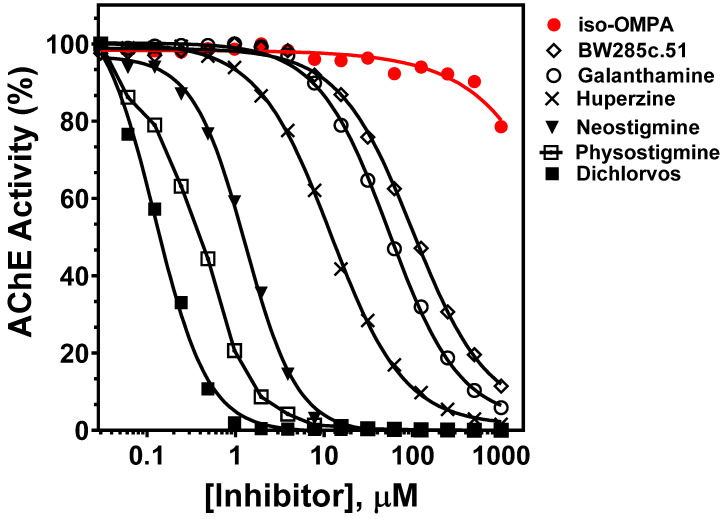
Inhibition of rSmTAChE by previously described AChE inhibitors. rSmAChE enzyme activity levels measured in the presence of increasing concentrations of the listed inhibitors. IC_50_ values calculated from these data are presented in Table 2. The butyrylcholinesterase-specific inhibitor (iso-OMPA, red) has minimal impact on rSmAChE activity and only at high concentration.

**Figure 3 ijms-26-01975-f003:**
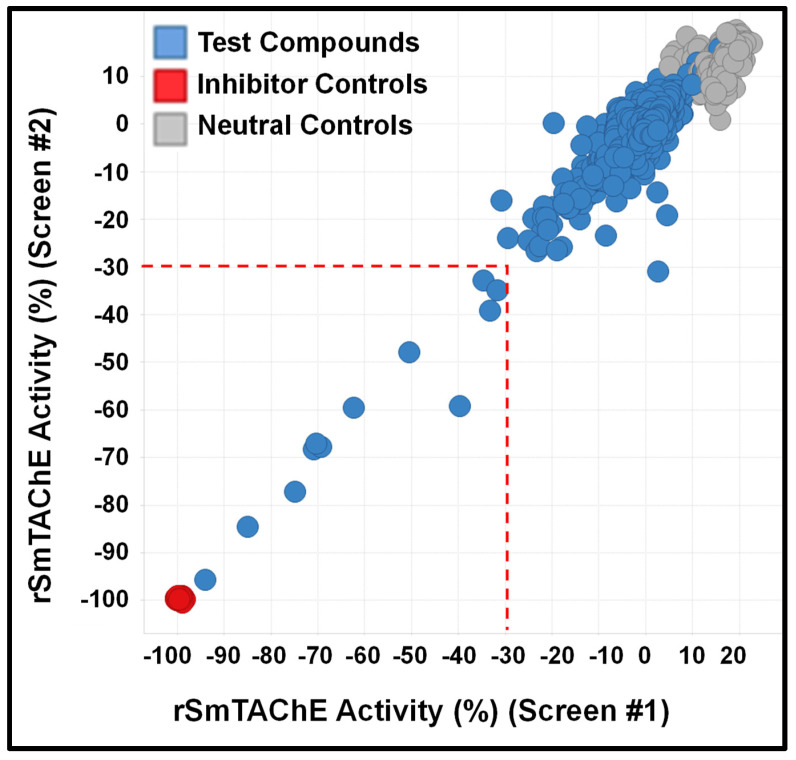
Chemical screen performance and reproducibility. Recombinant SmTAChE activity was monitored in the presence of test compounds (blue circles), and data are presented relative to 100% activity levels measured in the absence of any chemical (neutral controls, gray circles). Activity recorded in the absence of enzyme, equivalent to 0% activity (inhibitor controls, red circles). The screen was performed on two different days (screen #1 results are presented on the *x*-axis, and screen #2 results are presented on the *y*-axis). The two replicates, plotted against each other in the figure, show a linear fit of the data set, indicating good reproducibility. The red-dotted line encloses the 12 test compounds that reproducibly inhibit rSmTAChE by >30%.

**Figure 4 ijms-26-01975-f004:**
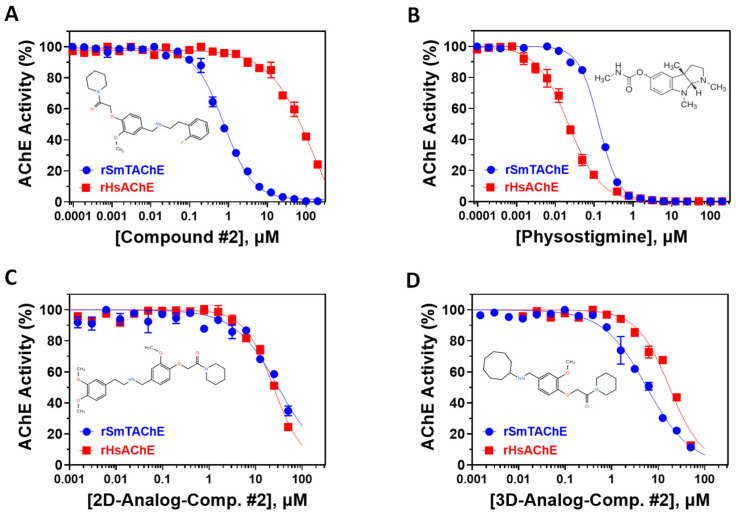
Specificity of SmTAChE inhibitors. The IC_50_ inhibition curves of compound #2 (**A**), as well as physostigmine (**B**), against rSmTAChE (blue) and rHsAChE (red) were generated by using GraphPad Prism. Similarly, the IC_50_ inhibition curves of compound #2’s 2D (**C**) and 3D (**D**) structural analogs are presented. The structure of each compound is depicted within its respective panel.

**Figure 5 ijms-26-01975-f005:**
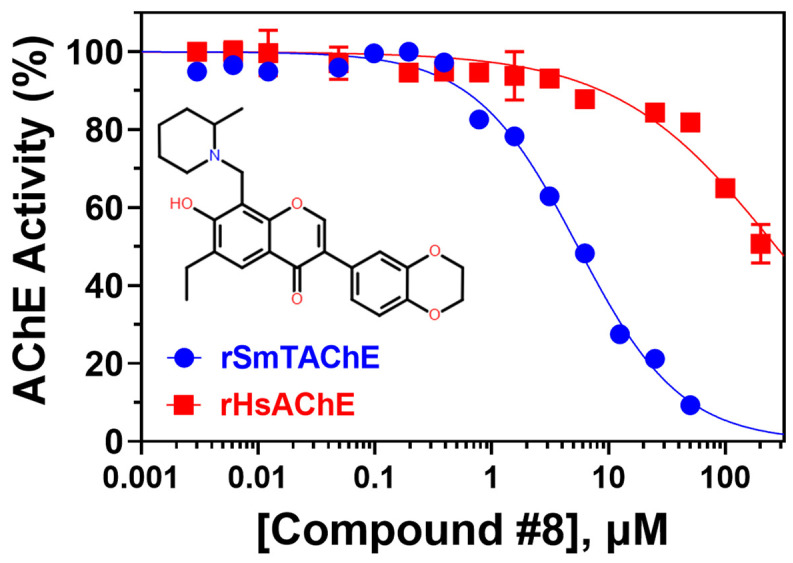
Specificity of compound #8. The IC_50_ inhibition curves of compound #8 against rSmTAChE (blue) and rHsAChE (red) were determined. The structure of compound #8 is depicted.

**Table 1 ijms-26-01975-t001:** rSmTAChE kinetics and substrate specificity.

Substrate	K_m_ (mM)	K_cat_ (S^−1^)	K_cat_/k_m_ (M^−1^·S^−1^) (×10^6^)
**ATCh**	0.32 ± 0.037	372.79 ± 38.6	1.17 ± 0.140
**BuTCh**	0.33 ± 0.060	10.67 ± 0.22	0.03 ± 0.005
**PTCh**	0.14 ± 0.002	74.16 ± 0.35	0.53 ± 0.004

**Table 2 ijms-26-01975-t002:** Inhibition of AChE activity (IC_50_ values, µM) using rSmTAChE and live schistosomula parasites.

	rSmTAChE (µM)	Schistosomula (µM)
Physostigmine	0.37	0.26
Neostigmine	1.29	0.8
Dichlorvos	0.12	0.2
Galanthamine	54.2	25
Huperzine	11.9	6
BW285c.51	101	32

**Table 3 ijms-26-01975-t003:** Major hits in the Broad Validation Library that reproducibly inhibited rSmTAChE. Inhibitors were used at 7.5 µM.

**#**	**Compound Broad ID** **(Structure)**	**Compound Name**	**Inhibition** **(%)**
**1**	**BRD-K25650355-059-14-8** 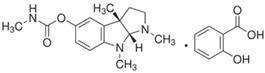	Physostigmine salicylate	94.90
**2**	**BRD-K77965629-003-12-9** 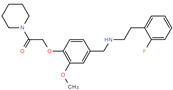	[2-(2-fluorophenyl)ethyl]{3-methoxy-4-[2-oxo-2-(1-piperidinyl) ethoxy]benzyl}amine hydrochloride	84.81
**3**	**BRD-K18922609-067-10-5** 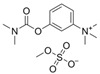	Neostigmine methyl sulfate	76.03
**4**	**BRD-K18922609-004-16-5** 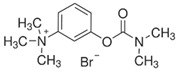	Neostigmine bromide	69.62
**5**	**BRD-K13183738-071-14-7** 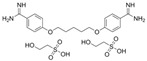	Pentamidine isethionate salt	68.69
**6**	**BRD-K99622919-001-17-2** 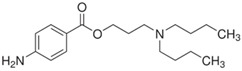	Butacaine; 149-16-6	68.57
**7**	**BRD-K50406511-001-18-7** 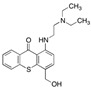	1-[(2-[diethylamino]ethyl)amino]-4-[hydroxymethyl]-9h-thioxanthen-9-one	60.91
**8**	**BRD-A35326728-001-11-3** 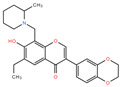	3-(2,3-dihydro-1,4-benzodioxin-6-yl)-6-ethyl-7-hydroxy-8-[(2-methyl-1-piperidinyl)methyl]-4H-chromen-4-one	49.19
**9**	**BRD-A38763545-001-12-3** 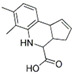	6,7-Dimethyl-3a,4,5,9b-tetrahydro-3H-cyclopenta[c]quinoline-4-carboxylic acid	49.44
**10**	**BRD-K49754828-001-11-7** 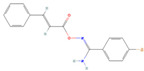	4-bromo-N′-(cinnamoyloxy) benzenecarboximidamide	33.74
**11**	**BRD-K12429625-001-11-4** 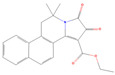	6,6-Dimethyl-8,9-dioxo-5,6,8,9-tetrahydro-benzo[f]pyrrolo [2,1-a] isoquinoline-10-carboxylic acid ethyl ester	36.17
**12**	**BRD-K24886212-001-12-8** 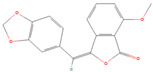	3-(1,3-benzodioxol-5-ylmethylene)-7-methoxy-2-benzofuran-1(3H)-one	33.23

## Data Availability

The data generated during this study are included in the published manuscript. The raw data supporting the conclusions of this article will be made available by the authors upon request.

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
