# Peer review of "Expression, Characterization and Selective Chemical Inhibition of Essential Schistosoma mansoni Tegumental Acetylcholinesterase (SmTAChE)"

_ijms, 2025, doi:10.3390/ijms26051975_

Round 1

Reviewer 1 Report

Comments and Suggestions for Authors

SmTAChE and human AChE are quite distinct sharing only 35% sequence identity, a springboard for discovering specific inhibitor, yet they are both sensitive to metrifonate and most if not all inhibitors, rendering the task of discovering a selective inhibitor for treating schistosomiasis quite arduous.

A very thorough and perfect work indeed, yet it is important to note that all assays and conclusions interpreted as due to enzymatic or other activity at the surface of live parasites have an important limitation:  they have nothing to do with the in vivo settings where parasites are unscathed by any technical procedure, striving in sterile blood (100% serum), surrounded by all the nutrients they need and like, impervious to immune effectors and cells, and lacking all the in vitro tests molecules, buffers, ingredients and additives, which more or less effectively impact the integrity of the surface double lipid bilayer cover.

Author Response

Dear Dr. Puriteerangkul,

Thank you and the reviewers for the assessments of our manuscript and the valuable comments. We are pleased with positive feedback from the reviewers. Here is our response to their minor comments (in bold):

Reviewer #1

SmTAChE and human AChE are quite distinct sharing only 35% sequence identity, a springboard for discovering specific inhibitor, yet they are both sensitive to metrifonate and most if not all inhibitors, rendering the task of discovering a selective inhibitor for treating schistosomiasis quite arduous.

Response:

The reviewer is correct that identifying parasite- specific inhibitors is typically difficult. However, we would like to highlight that in this initial screen, we were able to identify a potent parasite-specific inhibitor (Compound #2) with an IC50 value for rSmTAChE that is more than 200-fold lower than that for the human enzyme.

A very thorough and perfect work indeed, yet it is important to note that all assays and conclusions interpreted as due to enzymatic or other activity at the surface of live parasites have an important limitation:  they have nothing to do with the in vivo settings where parasites are unscathed by any technical procedure, striving in sterile blood (100% serum), surrounded by all the nutrients they need and like, impervious to immune effectors and cells, and lacking all the in vitro tests molecules, buffers, ingredients and additives, which more or less effectively impact the integrity of the surface double lipid bilayer cover.

Response:

Indeed, this is a limitation of all in vitro screens across all diseases. Once safe, specific, and effective hits are identified, they must be further validated for their effectiveness in vivo.

Reviewer 2 Report

Comments and Suggestions for Authors

The study titled "Expression, Characterization, and Selective Chemical Inhibition of the Essential Schistosoma mansoni Tegumental Acetylcholinesterase (SmTAChE)" by Skelly and Da’dara focuses on a non-neuronal acetylcholinesterase (AChE) from Schistosoma mansoni, named SmTAChE, as a potential drug target. The study thus addresses the urgent need for alternatives to praziquantel in treating schistosomiasis. The manuscript is well-presented and well-written, and I only have a few minor comments and suggestions to the authors before the publication of the study.

  • L12: S. mansoni = Schistosoma mansoni
  • L86-87: The sentence “Since several immune cells respond to acetylcholine, the enzyme may have an immunoregulatory role” could be improved to be more informative (i.e., specifying which immune cells and what type of response). Additionally, the authors might provide a brief explanation of acetylcholine-driven immune signaling in lines 87-89 to give the readers more context.
  • L106: The statement “Western blotting using anti-SmTAChE antibodies confirms more robust rSmTAChE production in cells transfected with the codon-optimized plasmid (arrow, lower panel, Supplementary Figure S1A)” seems overstated. The non-codon optimized version, labeled AChE in Supplementary Figure S1A, does not appear to be expressed at all based on the Western blots shown.
  • All SDS-PAGE and blots shown in Figure 1A should be included in the supplementary data as non-cropped figures.
  • Figure 2: The title of the X-axis needs to be corrected.
  • The final sequence for SmTAChE, optimized with hamster 386 codon preferences and cloned, should be presented in the supplementary data. The authors should mention that they have produced the non-codon optimized version of SmTAChE, as shown in their supplementary data.
  • L441: The number of replicates used and the experiments with the non-codon optimized version of SmTAChE should be included in item 4.5 of M and M.
  • L450: Butyrylthiocholine iodide (BuTCh, Sigma) or propionylthiocholine iodide (PTCh, Sigma) were tested as substrates. The concentrations used should be included.
  • L464-467: How many replicates were used? How long was the incubation period? Were the schistosomula monitored to ensure they were alive? What was the incubation temperature? L523: Similar questions.

  • Do the authors have any insight into how the inhibitors tested affect neural SmAChE?

  • L281-283: This is a strong statement. Did the authors try to correlate RNAi experiments to knock down rSmTAChE in live schistosomula with experiments using the key inhibitors identified in the present study? This could provide more confidence in this assertion.
  • L332: Why was Compound #9 not confirmed using a dose-response inhibition assay?

Author Response

Dear Dr. Puriteerangkul,

Thank you and the reviewers for the assessments of our manuscript and the valuable comments. We are pleased with positive feedback from the reviewers. Here is our response to their minor comments (in bold):

Reviewer #2

The study titled "Expression, Characterization, and Selective Chemical Inhibition of the Essential Schistosoma mansoni Tegumental Acetylcholinesterase (SmTAChE)" by Skelly and Da’dara focuses on a non-neuronal acetylcholinesterase (AChE) from Schistosoma mansoni, named SmTAChE, as a potential drug target. The study thus addresses the urgent need for alternatives to praziquantel in treating schistosomiasis. The manuscript is well-presented and well-written, and I only have a few minor comments and suggestions to the authors before the publication of the study.

  • L12: S. mansoni = Schistosoma mansoni

Response:

Corrected. Thanks

  • L86-87: The sentence “Since several immune cells respond to acetylcholine, the enzyme may have an immunoregulatory role” could be improved to be more informative (i.e., specifying which immune cells and what type of response). Additionally, the authors might provide a brief explanation of acetylcholine-driven immune signaling in lines 87-89 to give the readers more context.

Response:

The field of non-cholinergic acetylcholine signaling is rapidly advancing. Although it is not the primary focus of this manuscript, we have expanded on this topic in response to the reviewer’s request (lines 90-93)and included additional references (Ref. # 37-41).

  • L106: The statement “Western blotting using anti-SmTAChE antibodies confirms more robust rSmTAChE production in cells transfected with the codon-optimized plasmid (arrow, lower panel, Supplementary Figure S1A)” seems overstated. The non-codon optimized version, labeled AChE in Supplementary Figure S1A, does not appear to be expressed at all based on the Western blots shown.

Response:

We agree with the reviewer that western blot analysis does not show the expression of non-codon-optimized AChE. However, this is due to the sensitivity limitations of the technique. As shown in Figure S1, the enzyme activity assay clearly demonstrates that the non-codon optimized AChE is expressed indeed. Since the enzyme assay is more sensitive than western blot analysis, the absence of a signal in the blot does not indicate a complete lack of expression of the non-codon optimized AChE.

  • All SDS-PAGE and blots shown in Figure 1A should be included in the supplementary data as non-cropped figures.

Response:

No cropped figures are shown. We present the full-length SDS-PAGE and western blot. Other parts of the gels contain samples from unrelated projects, which are not relevant to this manuscript. The displayed images show the entire relevant section of the gel; no bands were cropped.

  • Figure 2: The title of the X-axis needs to be corrected.

Response:

The x-axis title is correct, representing the inhibitor concentrations in µM. It already displays the anti-log values, providing a more informative representation for readers.

  • The final sequence for SmTAChE, optimized with hamster codon preferences and cloned, should be presented in the supplementary data. The authors should mention that they have produced the non-codon optimized version of SmTAChE, as shown in their supplementary data.

 Response:

The requested information has been added as a supplementary Figure S3 and (lines 395-398).

  • L441: The number of replicates used and the experiments with the non-codon optimized version of SmTAChE should be included in item 4.5 of M and M.

Response:

This information has been added in section 4.5 and lines (462-466)

  • L450: Butyrylthiocholine iodide (BuTCh, Sigma) or propionylthiocholine iodide (PTCh, Sigma) were tested as substrates. The concentrations used should be included.

Response:

The concentrations used are all indicated in the text (line 471) as well as in figure 1B.

  • L464-467: How many replicates were used? How long was the incubation period? Were the schistosomula monitored to ensure they were alive? What was the incubation temperature? L523: Similar questions.

 Response:

We have included this information in the revised manuscript (Section 4.5; lines 481-485).

  • Do the authors have any insight into how the inhibitors tested affect neural SmAChE?

Response:

Unfortunately, we did not express neuronal AChE enzyme, and therefore, we can’t comment on the activity of these compounds on this enzyme.

  • L281-283: This is a strong statement. Did the authors try to correlate RNAi experiments to knock down rSmTAChE in live schistosomula with experiments using the key inhibitors identified in the present study? This could provide more confidence in this assertion.

 Response:

We respectfully disagree with the reviewer. Our statement simply asserts that inhibitors potent against rSmTAChE should also inhibit the native enzyme on the surface of that parasites. We have already demonstrated that rSmTAChE and native SmTAChE have similar kinetic properties. Furthermore, our data show that inhibitors effective against rSmTAChE are also effective against the native SmTAChE on the surface of the parasites (Table 2).

A comparative study of RNAi with chemical inhibition would indeed be informative; however, we have not yet conducted such experiments.

  • L332: Why was Compound #9 not confirmed using a dose-response inhibition assay?

Response:

In general, many screening hits fail to inhibit enzymes when analyzed through a dose-response curve. Compound #9 is one such compound. We tested compound #9 using a dose-response curve, but it did not inhibit the enzyme. This is what we meant by “not confirmed”. We have clarified this point in the revised manuscript (Lines 338-339)